

# Exploring HONO production from particulate nitrate photolysis in Chinese representative regions: characteristics, influencing factors and environmental implications

Bowen Li[1], Jian Gao[1], Chun Chen[1], Liang Wen[1], Yuechong Zhang[1], Junling Li[1], Yuzhe Zhang[1], Xiaohui Du[1], Kai Zhang[1], Jiaqi Wang[1]

[1]State Key Laboratory of Environmental Criteria and Risk Assessment, Chinese Research Academy of

Environmental Sciences, Beijing 100012, China

*Correspondence to: Jiaqi* Wang (wang.jiaqi@craes.org.cn), Kai Zhang (zhangkai@craes.org.cn)

**Abstract.** The production mechanism of atmospheric nitrous acid (HONO), an important precursor of

hydroxyl radical (OH), was still controversial. Few studies have explored the effects of particulate

nitrate photolysis on HONO sources in different environment conditions across China. Here, the

photolysis rate constants of particulate nitrate for HONO production ($J_{HONO}$) were determined through

photochemical reaction system with $PM_{2.5}$ samples collected from five representative sites in China. To

eliminate the "shadowing effect" — potential light extinction within aerosol layers at heavy $PM_{2.5}$

loadings on the filters, the relationship between light screening coefficient and EC, the dominant

light-absorbing component in $PM_{2.5}$, was established ($R^2$=0.73). The corrected $J_{HONO}$ values varied with

sampling period and location over a wide range, distributing from $1.6 \times 10^{-6}$ s$^{-1}$ to $1.96 \times 10^{-4}$ s$^{-1}$, with a

mean ($\pm$ 1 SD) of $(1.71 \pm 2.36) \times 10^{-5}$ s$^{-1}$. Chemical compositions, specifically nitrate loading and

organic component, affected the production of HONO through particulate nitrate photolysis: high $J_{HONO}$

values were generally associated with the $PM_{2.5}$ samples with high OC/NO$_3^-$ ratio ($R^2$=0.86). We

suggested that the parameterization equation between $J_{HONO}$ and OC/NO$_3^-$ established in this

work can be used to estimate $J_{HONO}$ in different aerosol chemical conditions, thus reducing the

uncertainty in exploring HONO daytime sources. This study confirms that the photolysis of

particulate nitrate can be a potential HONO daytime source in rural or southern urban sites,

which were characterized by high proportion of organic matter in $PM_{2.5}$, while the contribution

of this process to HONO daytime formation was still limited.



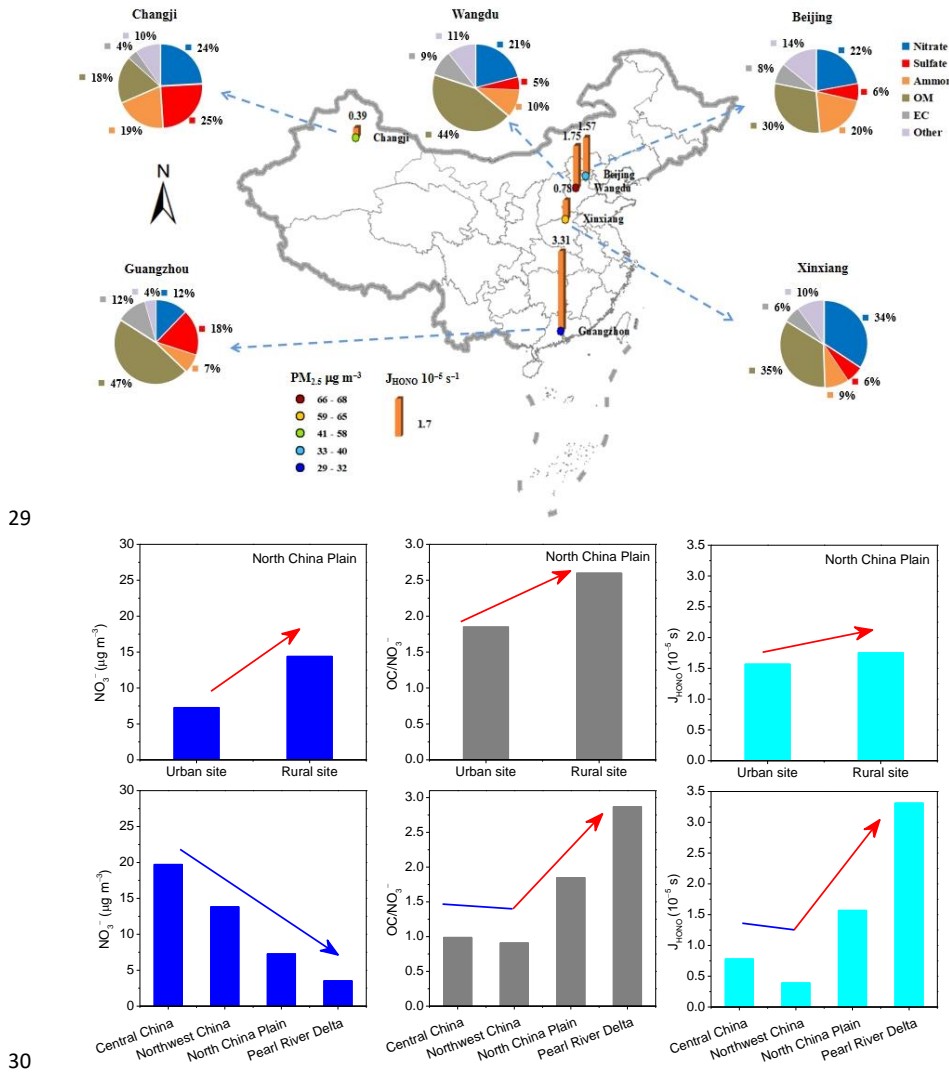



## 1 Introduction

Gaseous nitrous acid (HONO) is an important nitrogen-containing trace gas in the troposphere,
which can produce hydroxyl radical (OH) through photolysis, thus stimulating the enhancement of
atmospheric oxidation and the formation of secondary aerosols (Fu et al., 2019; Slater et al., 2020; Ren
et al., 2003; Li et al., 2011; Su et al., 2011). In recent years, the contribution of HONO to atmospheric
oxidation in heavily polluted conditions has attracted great attention (Villena et al., 2011; Fu et al.,
2019; Slater et al., 2020). Even though observational research on HONO has been conducted for nearly
40 years, the understanding of HONO daytime source was still controversial (Fu et al., 2019; Wang et
al., 2017; Mora Garcia et al., 2021). Numerous mechanisms have been proposed to explain the
extremely high HONO concentrations at noon, including direct combustion emission (Kurtenbach et al.,
2001; Liang et al., 2017; Liao et al., 2021), gas-phase reaction of NO and OH radical (Li et al., 2011;
Zhang et al., 2016), heterogeneous reaction of $NO_2$ (Wang et al., 2017; Ammann et al., 1998; Monge et
al., 2010; Stemmler et al., 2006), soil emissions (Su et al., 2011; Oswald et al., 2013; Melissa A, 2014;
Kim and Or, 2019), and the photolysis of $HNO_3$/nitrate on aerosol or ground surface (Zhou et al., 2003;
Zhou et al., 2011; Ye et al., 2016b; Ye et al., 2016a; Ye et al., 2017).
Particulate nitrate, which was conventionally considered as the ultimate oxidation product of $NO_x$,
can rapidly photolyze and recycle $NO_x$ or HONO back to the gas phase (Andersen et al., 2023; Handley
et al., 2007; Beine et al., 2006; Ye et al., 2016a; Ye et al., 2017; Ye et al., 2016b; Gu et al., 2022b), at a
rate 10 to 300 times faster than the photolysis rate of gaseous $HNO_3$ (~$7 \times 10^{-7}$ $s^{-1}$) under typical
tropical noontime conditions (Finlayson-Pitts, 2000). Recently, some field, laboratory and modeling
works have proposed that photolysis of particulate nitrate can be an important in situ source of HONO
in rural, suburban and urban environments (Ye et al., 2016b; Mora Garcia et al., 2021; Liu et al., 2019;
Bao et al., 2018; Wang et al., 2017). Fu et al. (2019) found that the photolysis of $HNO_3$/nitrate in the
atmosphere and deposited on surfaces was the dominant HONO source during noon and afternoon,
contributing above 50 % of the simulated HONO. However, there are large discrepancies in estimating
the rate constants in the atmosphere (Gen et al., 2022). In New York, Ye et al. (2017) reported that the
photolysis rates of particulate nitrate in clean areas were two orders of magnitude higher than that in
polluted areas, ranging from $6.2 \times 10^{-6}$ to $5.0 \times 10^{-4}$ $s^{-1}$, with a median of $8.3 \times 10^{-5}$ $s^{-1}$. The proposed rate
constants of nitrate photolysis based on the aircraft observations over South Korea ranged from $7 \times 10^{-6}$





to $2.1 \times 10^{-5}$ s$^{-1}$ (Romer et al., 2018). Shi et al. (2021) derived the rate constant ($< 2 \times 10^{-5}$ s$^{-1}$) based on
chamber experiments, but found a limited role of this mechanism to HONO production. The
uncertainty of HONO production rate from the photolysis of particulate nitrate can reach up to 1.4 ppbv
h$^{-1}$, and greatly affect the accuracy of HONO source analysis (Liu et al., 2019; Lee et al., 2016; Ye et
al., 2016a). The highly-varied photolysis rate constant of particulate nitrate was closely associated with
environmental conditions and the aerosol chemical or physical characteristics, such as relative humidity
(RH), aerosol acidity, light intensity, and coexisting components (organic components, halogen, etc.)
(Gelencsér et al., 2003; Ye et al., 2016a; Bao et al., 2020; Wang et al., 2021; Reeser et al., 2013). Thus,
elucidating the mechanism and dominant factors controlling the photolysis of particulate nitrate is
important to accurately estimate the contribution of this process to HONO daytime production.

In general, the photolysis rate constant of particulate nitrate was derived though photochemical

experiments using bulk particle samples collected on filters (Ye et al., 2017; Bao et al., 2018).
Comparing with the suspended particles in the ambient atmosphere, the collected PM$_{2.5}$ particles in the
aerosol filters may present a multiple-layer structure, especially in heavy air pollution conditions (Bao
et al., 2018). The light-absorbing species within PM$_{2.5}$ particles would hinder the light absorption of
particulate nitrate in the lower layers of the filter sample, thus inhibiting the photolysis of particulate
nitrate, which was called the "shadowing effect" (Ye et al., 2017). The shadowing effect may be
negligible in clean air conditions but should be evaluated and quantified in heavy haze conditions.
However, previous works generally ignored this shadowing effect.

According to previous field observations, the PM$_{2.5}$ chemical composition, especially particulate

nitrate concentration (NO$_3^-$), changed significantly across China (Wang et al., 2022a, b; Wang et al.,
2022c; Wang et al., 2016; Cheng et al., 2024). As one of the key industrial development areas in China,
the Pearl River Delta Region (PRD) has a great number of large-scale industrial parks dominated by the
chemical industry, resulting in significant VOC emissions and a large proportion of organic matter (OM)
in PM$_{2.5}$. In the North China Plain (NCP), the particulate nitrate (NO$_3^-$) has surpassed sulfate (SO$_4^{2-}$)
and OM to become the dominant PM$_{2.5}$ component in recent years (Wang et al., 2022b). For now, the
investigation of particulate nitrate photolysis in different atmospheric environments was limited in
China, and the influence of aerosol chemical or physical characteristics on HONO production was still
unclear. In this work, to shed light on the contribution of particulate nitrate photolysis to the HONO
daytime source, we examined the photolysis rate constant for HONO based on photochemical




experiments with $PM_{2.5}$ samples collected from five typical sites in China. In addition, the shadowing
effect due to increasing aerosol particle loading on the filters was quantified. After correcting this effect,
the influence of various environmental conditions, including particulate nitrate, organic matter, and
aerosol acidity, on the formation of HONO was investigated and the possible role of this photolytic
process as HONO sources was also examined.
**2 Method**
**2.1 Sampling and filter treatment**
The ambient $PM_{2.5}$ was collected on Teflon or quartz filters in autumn-winter seasons in five
representative sites, i.e., Beijing, Wangdu, Xinxiang, Guangzhou, and Changji, which were shown in
Figure 1a and described in detail in the Supporting Information. These cities were located in the North
China Plain (NCP, urban: Beijing, rural: Wangdu), Central China, Pearl River Delta Region (PRD), and
Northwestern China, respectively. The sampling flow rates ranged from 16.7 to 1050 L $min^{-1}$, the
sampling times from 9 h to 23 h, and the overall sampling volumes of air from 8 $m^3$ to 1450 $m^3$, to
collect a very wide range of particulate nitrate loadings. The comparison experiments between Teflon
and quartz filters have been conducted, and no significant differences in HONO production rates from
particulate nitrate photolysis have been found (T<0.01). The sampling settings employed in Wangdu
were designed to quantify the shadowing effect (Figure 1b). In Wangdu, $PM_{2.5}$ was collected at a flow
rate of 16.7 L $min^{-1}$ with four channels (A, B, C, and D). A and B channels were set for
daytime(8:00–17:00) and nighttime (18:00–7:00) $PM_{2.5}$ samples, respectively, and the other two
channels were for the "all-day" (including 8:00–17:00 and 18:00–7:00) $PM_{2.5}$ samples. A total of 158
effective $PM_{2.5}$ samples were obtained in this study. These aerosol filter samples were labeled and
stored at −20 ℃ in the freezer.
Fractions with given surface area from each filter sample were used to perform photochemical
reaction experiments and analysis of aerosol chemical components. For each $PM_{2.5}$ sample, the fraction
with given surface area was rinsed by deionized water and then sonicated for 15 min. The amounts of
water-soluble ions including $Na^+$, $NH_4^+$, $K^+$, $Mg^{2+}$, $Ca^{2+}$, $Cl^-$, $NO_3^-$, and $SO_4^{2-}$ were measured by ion
chromatography (IC, Thermo ICS-2100). To measure the values of carbon components, including
organic carbon (OC) and elemental carbon (EC), a part (0.5024 $cm^2$) of each filter was detected using a



thermal optical carbon analyzer (DRI model 2015). The concentration of OM was obtained by
multiplying the OC concentration by a factor of 1.6 (Li et al., 2021). $PM_{2.5}$ concentration was estimated
by the sum of all the water-soluble ions and carbon components. The surface concentration of $PM_{2.5}$
and its components on aerosol filters were calculated through dividing the absorbed loading with the
geometric area of the aerosol filter sample ($\mu g\ cm^{-2}$).

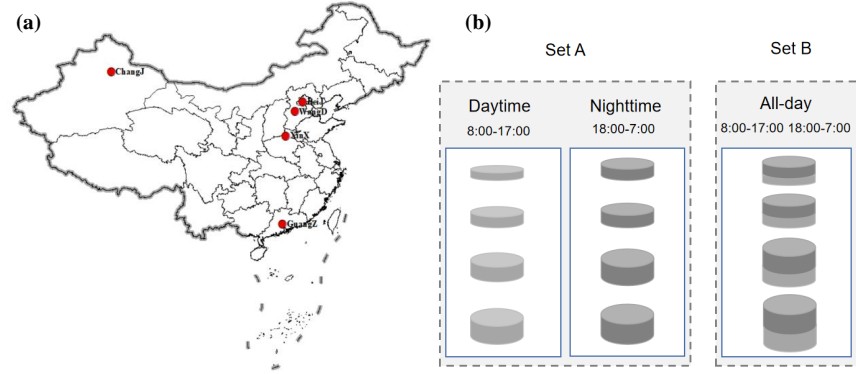


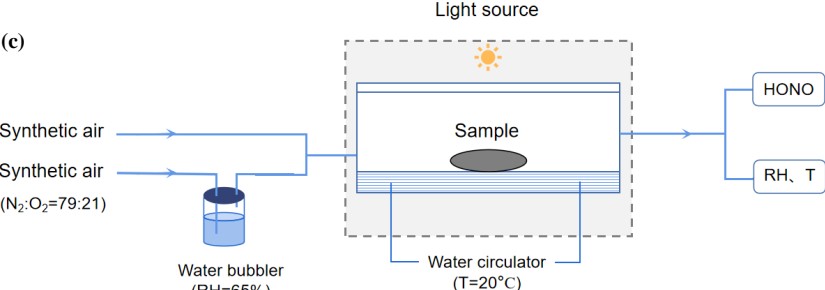


**Figure 1.** (a) Location map of five representative sampling sites in China, (b) the sampling settings to
quantify the shadowing effect in Wangdu, and (c) a schematic diagram of the photochemical
experimental setup.
**2.2 Photochemical reaction system**
A custom-made cylindrical quartz vessel was used as the photochemical flow reactor (Figure 1c).
The diameter was 10 cm and the depth was 2.5 cm, with a cell volume of ~200 ml. A xenon lamp (300
W) was placed 20 cm above the reactor as the light source. The light was filtered by a Pyrex sleeve to
remove heat-generating infrared light. The effective light intensity in the center of the flow reactor,
where aerosol samples were placed, was measured to be about 0.5 times higher ($1.5\ kW\ m^{-2}$, measured



by a calibrated optical power meter) than that at tropical noon on the ground (solar elevation angle
θ=0 °). Synthetic air, composed of ultrahigh-purity nitrogen and ultrahigh-purity oxygen mixed at a
ratio of 79:21, was used as the carrier gas. The relative humidity (RH) in the air flow was adjusted
through a water bubbler and monitored with an online RH sensor (Vaisala, HMT130). The aerosol filter
sample was exposed to the solar simulator radiation for 20 min. The photochemical reaction
experiment for each sample was repeated 2−3 times with different fractions from the same sample. The
gaseous product (i.e., HONO) released during the experiment was flushed out of the reactor by the
carrier gas and was detected online by a custom-built HONO analyzer, which had been applied in
several measurements previously (Zhang et al., 2020b; Li et al., 2021).
**2.3 HONO Production from the photolysis of particulate nitrate**

The production rates (nmol h$^{-1}$) of HONO from particulate nitrate photolysis ($P_{HONO}$) were

calculated from their time-integrated signals above the baselines over the period of light exposure:
$$P_{HONO} = \frac{F_g \times 60}{V_m(t_2-t_1)} \int_{t_1}^{t_2} C_{HONO} dt \qquad (1)$$
Where $F_g$ (L min$^{-1}$) is the flow rate of the carrier gas, $V_m$ (24.5 L mol$^{-1}$) is the molar volume of gas at
25 ℃ and 1 atm of pressure; $t_1$ and $t_2$ (min) are the starting and ending time of the irradiation,
respectively; $C_{HONO}$ (ppb) is the online measured concentration of HONO. With the flow rate of 2.5 L
min$^{-1}$, the residence time in the reaction system was around ~5 s. The photolytic loss of HONO was
less than 5 %, thus no correction was made in the calculation of HONO production.

The photolysis rate constant of particulate nitrate leading to HONO production ($J_{HONO}$, s$^{-1}$) was

calculated by the following equation:
$$J_{HONO} = \frac{P_{HONO}}{N_{NO_3^-} \times 3600} \qquad (2)$$
Where $N_{NO_3^-}$ (mol) is the amount of $NO_3^-$ in the tested PM$_{2.5}$ sample. In principle, the photolysis rate
constant should be calculated on the amount of $NO_3^-$ that is reachable to the irradiation. However, the
amount of the light-reachable $NO_3^-$ in the PM$_{2.5}$ sample was hard to quantify. In this work, the
deviation of $J_{HONO}$ due to the overestimate of the amount of $NO_3^-$ under light irradiation, which was
called the shadowing effect, would be corrected in Sect. 3.1.
**3 Results**





### 3.1 Quantify the influence of the shadowing effect

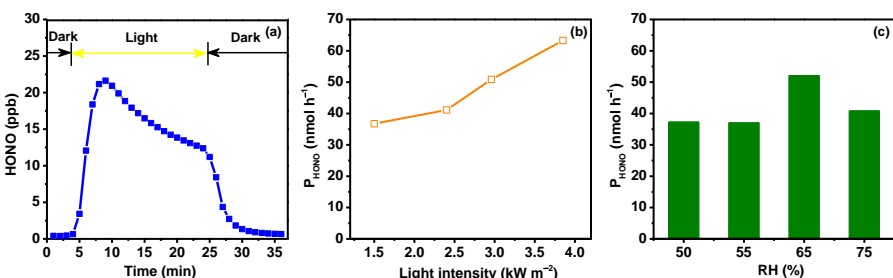

**Figure 2.** (a) Online measured concentrations of HONO during the light-exposure of an aerosol sample collected on June 12, 2023 in Beijing, $P_{HONO}$ as a function of (b) light intensity (kW m$^{-2}$) and (c) RH (%).

HONO production within the first 20 min of irradiation during the photochemical experiment was investigated on the PM$_{2.5}$ samples collected from five typical sites in China. Figure 2a showed a typical profile of the changes in HONO concentration in the reaction system. When the light was turned on, HONO concentration in the reactor increased immediately, then leveled off and slightly decayed afterwards. After the light was turned off, the HONO generation stopped immediately and the signal nearly returned to the baseline level. Previous works have revealed that the decay of HONO generation during light exposure period was not resulted from the evaporation loss of particulate nitrate (Ye et al., 2017), but mainly related to the inhomogeneity of particulate nitrate photochemical reactivity or the consumption of reactive electron donors, such as acidic proton (Bao et al., 2018). HONO production from the photochemical reactions of particulate nitrate were significantly influenced by ambient environmental conditions (i.e., light intensity and RH). As shown in Figure 2b, with the increase of light intensity, $P_{HONO}$ gradually increased, with $P_{HONO}$ in 3.85 kW m$^{-2}$ approximately twice than that in 1.50 kW m$^{-2}$. Previous works found that the formation of HONO was negligible at low RH (<5%), and increased at intermediate RH (15%−75%), then turned to decrease at RH > 90% (Bao et al., 2018). Here, we found that $P_{HONO}$ climbed to its highest when RH was around 65 % (Figure 2c). In this work, the photochemical reactions on different aerosol samples were all conducted under the same environmental condition (RH=65 %, temperature=20 ℃, and light intensity=1.50 kW m$^{-2}$).





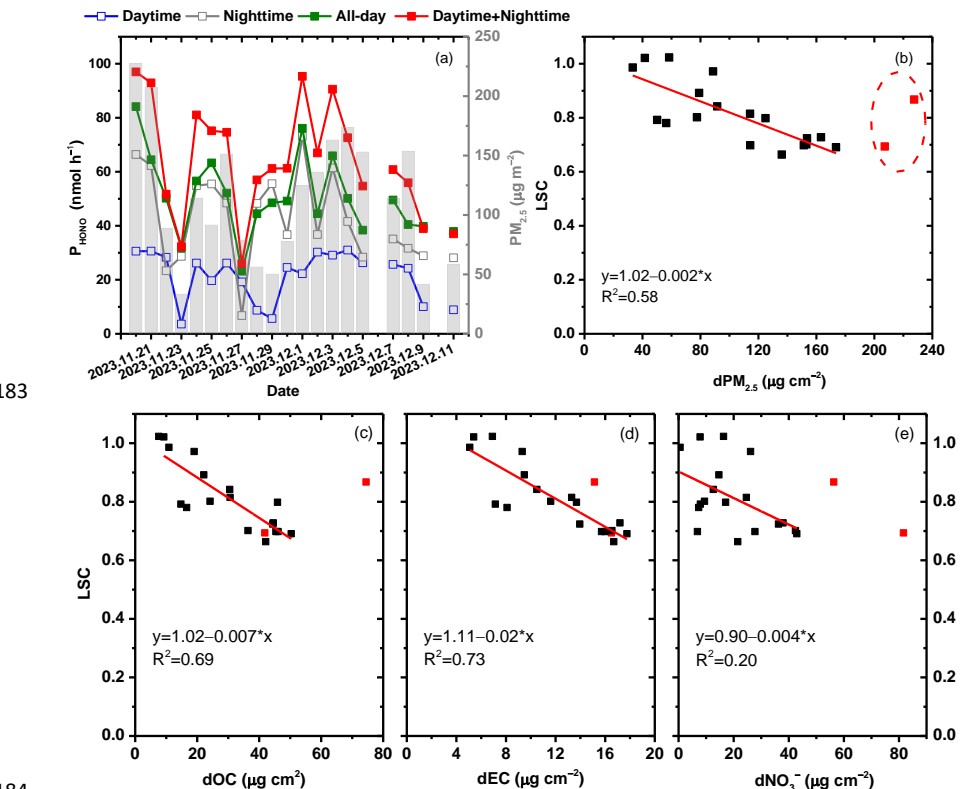

**Figure 3.** (a) Temporal variation of $P_{HONO}$ for aerosol filters collected in Wangdu during daytime, nighttime and all-day from November 20, 2023 to December 11, 2023, (b)-(e) relationships between light screening coefficient (LSC) and the surface concentrations of $PM_{2.5}$ ($dPM_{2.5}$), OC (dOC), EC (dEC) and $NO_3^-$ ($dNO_3^-$), respectively. The red squares represent the aerosol samples with $PM_{2.5}$ surface concentration higher than 200 μg cm$^{-2}$.

As expected, $P_{HONO}$ increased with particulate nitrate loadings in different sampling locations (Figure S1), however, it's interesting to note that, $P_{HONO}$ dose not increase or somewhat decrease at very high $NO_3^-$ loading condition. Previous works considered this may be attributed to the shadowing effect, wherein the particulate nitrate underneath the aerosol filters may receive less UV light at heavy aerosol particle loading on the filters, inhibiting the photolysis of particulate nitrate (Ye et al., 2017). Thus, the reported $P_{HONO}$ values would be underestimated under polluted ambient conditions. To verify and quantify the underestimation of $P_{HONO}$ due to the shadowing effect, we collected two sets of filters in Wangdu (set A: daytime and nighttime, set B: all-day, Figure 1b). Theoretically, the all-day one should share the same $NO_3^-$ loading and chemical composition as the sum of the daytime and nighttime filters,





thus the sum of $P_{HONO}$ during daytime ($P_{daytime}^{HONO}$) and nighttime ($P_{nighttime}^{HONO}$) should be equal to that
during all-day ($P_{all-day}^{HONO}$) without considering the shadowing effect. A total of 20 pairs of comparative
photochemical experiments were conducted, and the comparison of $P_{HONO}$ between these two sets of
filters was shown in Figure 3a. We found that the discrepancy between $P_{all-day}^{HONO}$ and $P_{daytime}^{HONO}$ +
$P_{nighttime}^{HONO}$ was widening along with the increase of surface $PM_{2.5}$ concentration. To quantify the
shadowing effect, we introduced a parameter called "light screening coefficient" (LSC) to describe the
decreasing efficiency of light penetrating into the particle with increasing $PM_{2.5}$ loadings:
$P_{theory}^{HONO} = P_{daytime}^{HONO} + P_{nighttime}^{HONO}$ (3)
$LSC = P_{observed}^{HONO}/P_{corrected}^{HONO} = P_{all-day}^{HONO}/P_{theory}^{HONO}$ (4)
As shown in Figure 3b, when $PM_{2.5}$ surface concentration ($dPM_{2.5}$) was low, LSC was almost
equal to 1, indicating that the shadowing effect was negligible. With the increase of $PM_{2.5}$ loading, the
value of LSC declined to lower than 65 %. In general, significant negative correlation exited between
LSC and $dPM_{2.5}$, except when $dPM_{2.5}$ was higher than 200 μg cm$^{-2}$ (Figure 3b). In this experiment, we
assumed that the daytime and nighttime $PM_{2.5}$ samples were both single-layered. However, with the
increase of air pollution, these filters in each pair of comparative experiments may already have
exhibited the shadowing effect, thus the sum of $P_{daytime}^{HONO}$ and $P_{nighttime}^{HONO}$ would be underestimated.
Therefore, when quantifying the shadowing effect, the LSC data with $PM_{2.5}$ loading higher than 200 μg
cm$^{-2}$ was excluded. Correlations between LSC and the surface concentrations of $PM_{2.5}$ major chemical
components, such as EC (dEC), OC (dOC), and $NO_3^-$ ($dNO_3^-$), were conducted (Figure 3c-e).
Significant correlation was found between LSC and carbonaceous component, especially EC ($R^2$=0.73),
which was one of the most important light absorbing species in $PM_{2.5}$, indicating that the shadowing
effect was mainly related to the light absorption components in $PM_{2.5}$. The relationship between LSC
and dEC was established as following:
dEC > 5.5 μg m$^{-2}$: LSC = 1.11−0.02×dEC
dEC ≤ 5.5 μg m$^{-2}$: LSC = 1 (5)
when dEC ≤ 5.5 μg m$^{-2}$, the shadowing effect can be ignored; when dEC > 5.5 μg m$^{-2}$, $P_{HONO}$ can be
corrected by the observed $P_{HONO}$ and LSC, which was estimated using this fitting equation with dEC.
Previous works found that the heavy loads of carbonaceous particles can turn these filters into dark
brown colors. The UV light was unlikely to transmit efficiently through the dark layer to the particulate
nitrate underneath, thus inhibiting the generation of HONO from the photolysis of particulate nitrate



(Ye et al., 2017). In consideration of the potential shadowing effect for the daytime and nighttime filters in each pair of comparative experiments, the $P_{daytime}^{HONO}$ and $P_{nighttime}^{HONO}$ observed would be underestimated, and the uncertainty of LSC should be considered at high $PM_{2.5}$ loadings. To evaluate this uncertainty, the observed $P_{daytime}^{HONO}$ and $P_{nighttime}^{HONO}$ values were recalculated and corrected to the theoretical single-layered condition based on Eq. (4) and (5). As shown in Figure S2, with the increase of $PM_{2.5}$ surface concentration, the deviations between LSC and the corrected one have enlarged. However, it's noted that the deviation was still lower than 20 % when $PM_{2.5}$ surface concentration was around 200 μg cm$^{-2}$. For example, for the aerosol sample collected in December 4, 2023, in Wangdu, the $PM_{2.5}$ surface concentration was 173.57 μg cm$^{-2}$, and the deviation was 15.74 %, which was acceptable in this work.

**3.2 Spatial distribution and temporal variation of HONO production from particulate nitrate photolysis**

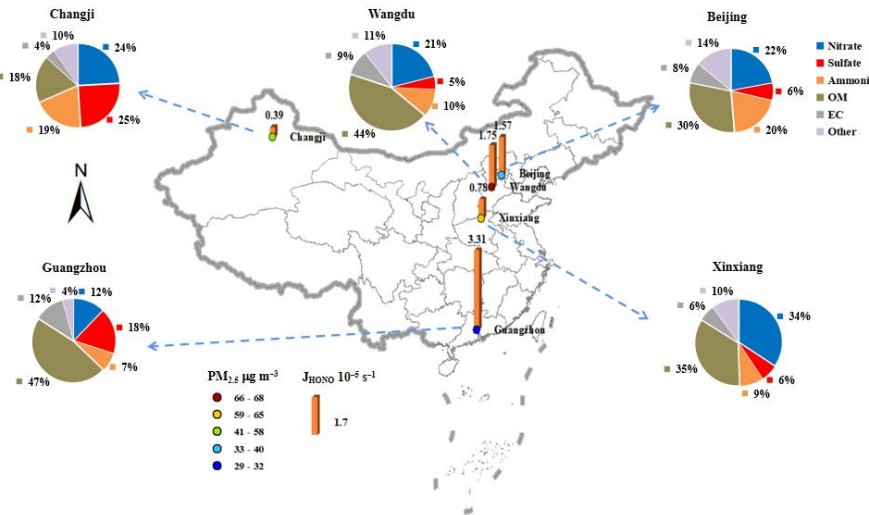

**Figure 4.** Spatial distribution of the average $J_{HONO}$, $PM_{2.5}$ loading, and chemical composition of the aerosol filters collected from five representative cities in China during the observation period.

There were 158 filter samples collected from five representative cities in China, and the averaged concentrations of $PM_{2.5}$ and its chemical composition of these filters showed significant spatial characteristics as shown in Figure 4. During the sampling period, OM was the most abundant species in $PM_{2.5}$ over most regions, except in the northwestern city (Changji), and $NO_3^-$ was the dominant inorganic component in the NCP (Beijing and Wangdu) and Central China (Xinxiang), while $SO_4^{2-}$





showed the highest contribution in the PRD (GuangZ) and Northwestern China (Changji). The values
of $J_{HONO}$ on these $PM_{2.5}$ samples were calculated by Eq. (2) with the $P_{HONO}$ corrected by Eq. (4) and (5),
and summarized in Figure 4 and Table 1. The corrected $J_{HONO}$, median and mean ($\pm$ one standard
deviation), were $1.55 \times 10^{-5}$ $s^{-1}$ and $1.57$ ($\pm 2.14$) $\times 10^{-5}$ $s^{-1}$ in Beijing, $1.68 \times 10^{-5}$ $s^{-1}$ and $1.75$ ($\pm 2.83$)
$\times 10^{-5}$ $s^{-1}$ in Wangdu, $0.69 \times 10^{-5}$ $s^{-1}$ and $0.78$ ($\pm 0.48$) $\times 10^{-5}$ $s^{-1}$ in Xinxiang, $3.04 \times 10^{-5}$ $s^{-1}$ and $3.31$
($\pm 1.15$) $\times 10^{-5}$ $s^{-1}$ in Guangzhou, and $0.38 \times 10^{-5}$ $s^{-1}$ and $0.39$ ($\pm 0.25$) $\times 10^{-5}$ $s^{-1}$ in Changji, respectively.
The maximum $J_{HONO}$ in these cities ranged from $0.91 \times 10^{-5}$ $s^{-1}$ in Changji to $1.96 \times 10^{-4}$ $s^{-1}$ in Wangdu.
These values were in the comparable range to those previously reported for aerosol samples, such as
$1.22 \times 10^{-5}$ $s^{-1} \sim 4.84 \times 10^{-4}$ $s^{-1}$ in China by Bao et al. (2018) and $6.2 \times 10^{-6}$ to $5.0 \times 10^{-4}$ $s^{-1}$ in US by Ye et
al. (2017). It's interesting to note that the average $J_{HONO}$ was the highest in Guangzhou, which was
characterized with the lowest $PM_{2.5}$ and $NO_3^-$ concentration among these cities. As for other cities with
high $PM_{2.5}$ concentrations, such as Changji and Xinxiang, the corrected $J_{HONO}$ was comparatively lower.
According to the National Ambient Air Quality Standard of China (GB3095-2012), the daily $PM_{2.5}$
averages in Guangzhou can meet the Level II standard of 75 $\mu g\,m^{-3}$, while exceeding the level I
standard (35 $\mu g\,m^{-3}$). Here, we defined $PM_{2.5}$ polluted days with daily mean $PM_{2.5}$ exceeding 35 $\mu g\,m^{-3}$.
As shown in Figure 5, the distribution of the corrected $J_{HONO}$ values in clean days were generally more
dispersed and higher than those in polluted days, except in Guangzhou. The average value of $J_{HONO}$ in
Guangzhou during air polluted conditions was slightly higher than that in clean conditions, besides
much higher than the values in other cities. Because the influence of the shadowing effect has been
corrected to some degree, these spatial and temporal change characteristics of $J_{HONO}$ in this work should
be mainly related to the varied chemical and physical properties of $PM_{2.5}$ samples collected from
different atmospheric environments.
**Table 1.** The concentrations of $PM_{2.5}$ chemical composition, corrected $J_{HONO}$, and $S_{HONO}$ in five
representative cities in China under different air conditions during the sampling period.

| Site | Air condition | $PM_{2.5}$ ($\mu g\,m^{-3}$) | $NO_3^-$ ($\mu g\,m^{-3}$) | OC ($\mu g\,m^{-3}$) | $OC/NO_3^-$ | Corrected $J_{HONO}$ ($10^{-5}$ $s^{-1}$) [a] | $S_{HONO}$ ($10^{-5}$ mol $h^{-1}$ $m^{-2}$) [b] | $S_{HONO}$ (ppbv $h^{-1}$) [c] |
|---|---|---|---|---|---|---|---|---|
|  | Clean | 19.71 | 3.15 | 3.89 | 2.25 | 2.01 | 0.15 | 0.03 |
|  | Polluted | 72.56 | 19.71 | 12.62 | 0.87 | 0.61 | 0.38 | 0.09 |
| Beijing | Whole-Min | 4.32 | 0.08 | 1.07 | 0.32 | 0.21 | 0.04 | 0.01 |
|  | Whole-Max | 102.64 | 32.90 | 15.95 | 12.82 | 11.06 | 0.57 | 0.13 |
|  | Whole-Mean | 32.92 | 7.29 | 6.07 | 1.85 | 1.57 | 0.22 | 0.05 |





|  |  |  |  |  |  |  |  |  |
|---|---|---|---|---|---|---|---|---|
|  | Clean | 20.39 | 3.05 | 3.61 | 1.66 | 0.65 | 0.07 | 0.02 |
|  | Polluted | 80.49 | 20.59 | 8.35 | 0.44 | 0.21 | 0.16 | 0.04 |
| Changji | Whole-Min | 14.45 | 0.88 | 2.69 | 0.28 | 0.16 | 0.03[d] | 0.01[d] |
|  | Whole-Max | 169.35 | 28.28 | 14.34 | 3.65 | 0.91 | 0.22 | 0.05 |
|  | Whole-Mean | 57.37 | 13.84 | 6.53 | 0.91 | 0.39 | 0.13 | 0.03 |
|  | Clean | 25.62 | 3.29 | 6.89 | 2.72 | 3.25 | 0.36 | 0.08 |
|  | Polluted | 40.32 | 4.38 | 13.82 | 3.35 | 3.53 | 0.59 | 0.13 |
| Guangzhou | Whole-Min | 14.77 | 0.85 | 3.67 | 0.82 | 1.37 | 0.17 | 0.04 |
|  | Whole-Max | 42.74 | 6.63 | 15.62 | 8.05 | 5.83 | 0.75 | 0.17 |
|  | Whole-Mean | 29.12 | 3.55 | 8.54 | 2.87 | 3.31 | 0.41 | 0.09 |
|  | Clean | 22.16 | 3.29 | 5.36 | 4.79 | 3.80 | 0.20 | 0.04 |
|  | Polluted | 83.53 | 18.06 | 23.23 | 1.88 | 1.09 | 0.50 | 0.11 |
| Wangdu | Whole-Min | 10.67 | 0.24 | 2.72 | 0.22 | 0.23 | 0.06 | 0.01 |
|  | Whole-Max | 173.45 | 60.28 | 63.07 | 22.06 | 19.60 | 0.88[e] | 0.20[e] |
|  | Whole-Mean | 68.38 | 14.41 | 18.82 | 2.60 | 1.75 | 0.42 | 0.10 |
|  | Clean | 23.53 | 4.35 | 5.69 | 1.37 | 1.28 | 0.21 | 0.05 |
|  | Polluted | 68.98 | 24.87 | 14.63 | 0.87 | 0.62 | 0.40 | 0.09 |
| Xinxiang | Whole-Min | 18.32 | 2.37 | 2.33 | 0.30 | 0.19 | 0.09 | 0.02 |
|  | Whole-Max | 143.10 | 73.47 | 22.06 | 2.02 | 1.96 | 0.59 | 0.13 |
|  | Whole-Mean | 57.62 | 19.74 | 12.40 | 0.99 | 0.78 | 0.35 | 0.08 |

[a] represented the photolysis rate constant of particulate nitrate leading to HONO production after considering the
influence of the shadowing effect. [b, c] represented the noontime source strength of HONO through the photolysis of
particulate nitrate with the units of $10^{-5}$ mol h$^{-1}$ m$^{-2}$ and ppbv h$^{-1}$, respectively.[d, e] represented the minimum and
maximum values of $S_{HONO}$ during the observation period.

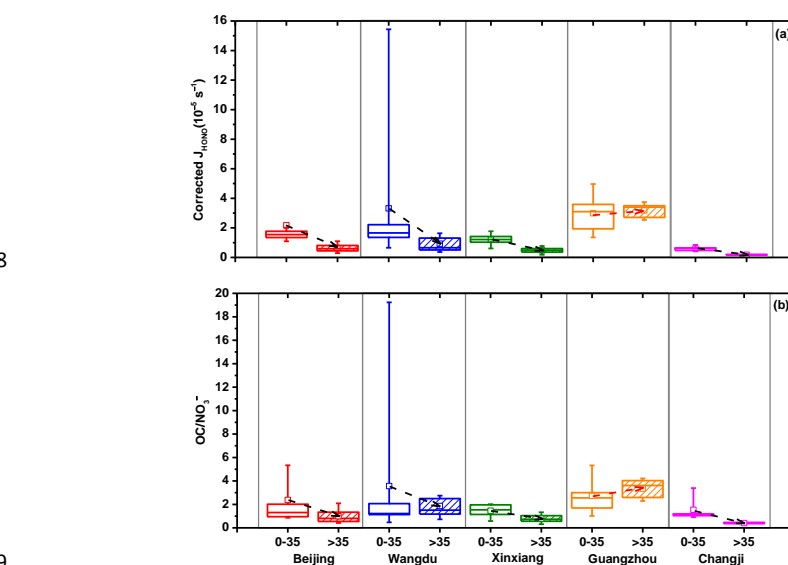






**Figure 5.** (a) Average corrected $J_{HONO}$, and (b) the ratio of OC to $NO_3^-$ under different air conditions in five representative cities. The box represents the 25th to 75th percentiles, the horizon line represents the median, the hollow square represents the mean, and the 10th and the 90th percentiles are the bottom and top whiskers, respectively.

## 3.3 Dominant factors controlling $J_{HONO}$

### 3.3.1 Particulate nitrate

As shown in Table 1, the corrected $J_{HONO}$ values varied with sampling periods and locations over a wide range, distributing from $0.16 \times 10^{-5}$ $s^{-1}$ for the aerosol sample collected in Changji with $PM_{2.5}$ higher than 90 $\mu g$ $m^{-3}$, to $19.60 \times 10^{-5}$ $s^{-1}$ for the aerosol sample collected in Wangdu with $PM_{2.5}$ lower than 25 $\mu g$ $m^{-3}$. Several factors may contribute to the discrepancy of $J_{HONO}$ in these different aerosol samples, such as particulate nitrate, organic matter, and aerosol acidity.

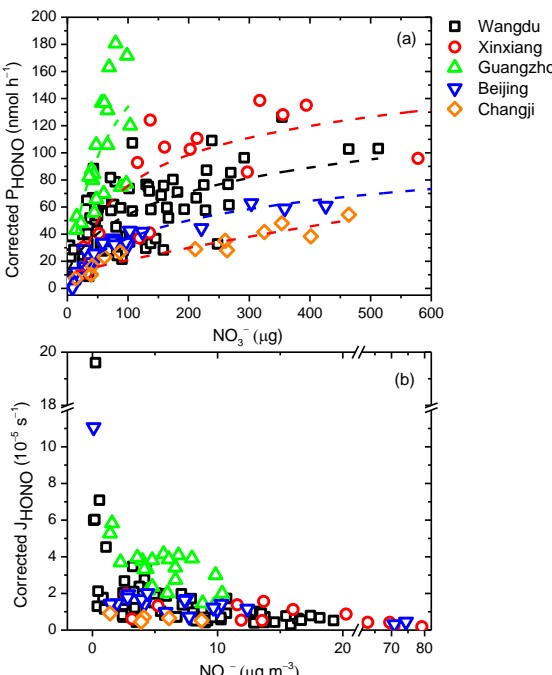

**Figure 6.** Relationships between (a) corrected $P_{HONO}$ and particulate nitrate loading, and (b) corrected $J_{HONO}$ and particulate nitrate concentration in different sampling locations. The dash lines in (a) were the best fits to the data for the fitting equation: the aerosol samples in Guangzhou (a=4.30, b=0.06, c=1 $\times 10^{-6}$, $R^2$= 0.42), Wangdu (a=2.54, b=0.11, c=1 $\times 10^{-6}$, $R^2$=0.50), Beijing (a=1.51, b=0.06, c=1$\times 10^{-6}$,





$R^2$=0.91), Xinxiang (a=2.28, b=0.06, c=1×10$^{-6}$, $R^2$=0.47), and Changji (a=0.58, b=0.04, c=1×10$^{-6}$,
$R^2$=0.86).
As shown in Figure 6, after considering the shadowing effect, the corrected $P_{HONO}$ generally
increased along with the increased amount of particulate nitrate (pNO$_3^-$, μg), but still gradually slowed
down at high particulate nitrate loading, resulting in a rapid decrease in $J_{HONO}$. For example, when
NO$_3^-$ concentration was at low level (around 0.5 μg m$^{-3}$) in Wangdu, the value of corrected $J_{HONO}$ was
about 30 times higher than that at high NO$_3^-$ concentration (around 20 μg m$^{-3}$). Previous works found
that the particulate nitrate was associated with matrix components in aerosol samples, and the
photolysis reactivity of particulate nitrate was closely associated with the surface catalysis effect (Ye et
al., 2017). In such a mechanism, the interaction between particulate nitrate and the substrate can distort
the molecular structure of nitrate and increase the absorption cross-section. The increases of $P_{HONO}$ with
pNO$_3^-$ exposed to the light radiation can be fitted by a logarithm curve under different
environment:$P_{HONO} = \frac{a}{b}\ln(1 + b(pNO_3^-)) + c(pNO_3^-)$ (Ye et al., 2017; Ye et al., 2019). Based on this
fitting equation, the corrected $P_{HONO}$ as a function of pNO$_3^-$ was showed in Figure 6a. Interestingly,
these relationships under different sampling locations showed distinct upward trends. Ye et al. (2019)
found that this ratio of a to b was related to the catalysis power of surface reactive sites and the organic
matters in the matrix. The much higher ratio of a (4.30) to b (0.06) values fitted for Guangzhou than
those for other cities, especially Changji (a=0.58, b=0.04), suggested extra catalytic power of organic
components in addition to the surface reactive site on particulate nitrate. The large deviation of the ratio
of a to b among these cities indicated the limitation of predicting $P_{HONO}$ only based on the relationship
with particulate nitrate in different atmospheric environments, and other varied aerosol chemical and
physical conditions should be considered as well.
**3.3.2 Organic matter**

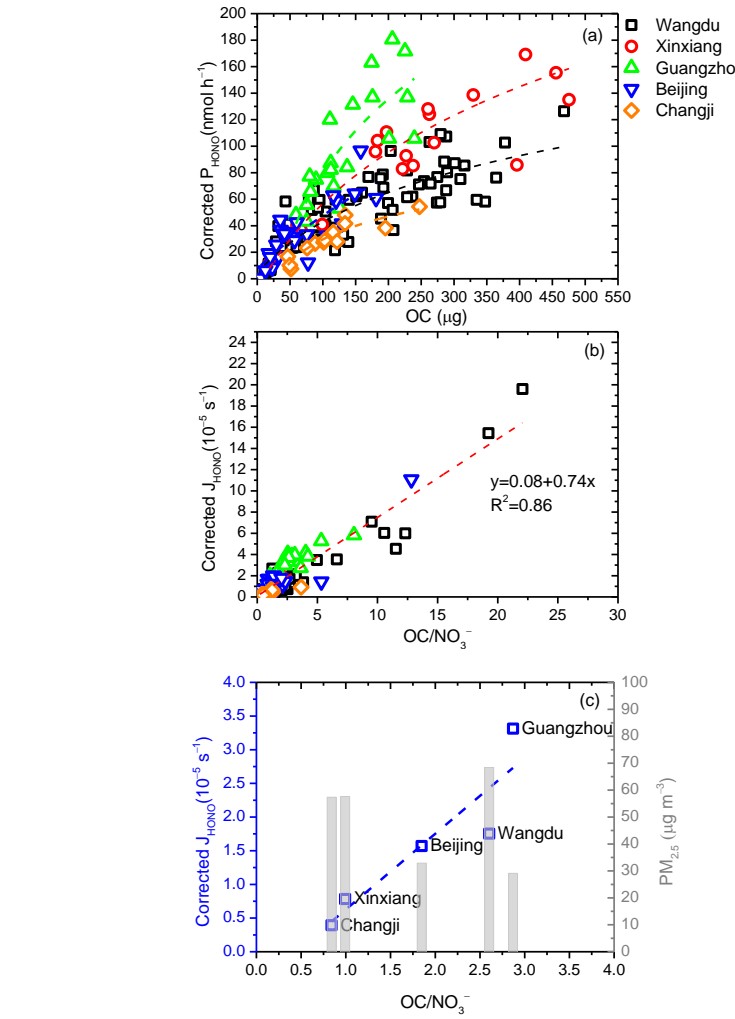



**Figure 7.** Relationship between (a) corrected $P_{HONO}$ and OC loadings, (b) corrected $J_{HONO}$ and OC/NO$_3^-$,

and (c) average corrected $J_{HONO}$, PM$_{2.5}$, and OC/NO$_3^-$ during the sampling period in five representative

cities.

Organic matter was ubiquitous in the atmosphere and contributed significantly to the total aerosol

mass. The selectivity of organic matter that coexisted in the aerosols was very important for the

production of HONO from the photolysis of particulate nitrate (Bao et al., 2018; Ye et al., 2016a;

Svoboda et al., 2013; Reeser et al., 2013; Stemmler et al., 2006; Yang et al., 2018; Beine et al., 2006;

Wang et al., 2021). As shown in Figure 7a, corrected $P_{HONO}$ generally increased as the amount of OC in

aerosol samples (pOC, μg) went up, while these positive correlations between $P_{HONO}$ and pOC shown



may be due to the moderate correlation between $pNO_3^-$ and $pOC$ ($R^2=0.39$, Figure S3). To eliminate
the contribution from particulate nitrate, the dependence of $J_{HONO}$ on the ratio of OC to $NO_3^-$ ($OC/NO_3^-$)
was examined:
Corrected $J_{HONO}=0.74\times(OC/NO_3^-)+0.08$          (6)
As shown in Figure 7b, significant linear correlation between corrected $J_{HONO}$ and $OC/NO_3^-$ was
found, with an $R^2$ of 0.86. In general, high corrected $J_{HONO}$ values were mostly associated with high
$OC/NO_3^-$ ratios for aerosol samples collected in the clean areas, such as Guangzhou, where the
averaged $PM_{2.5}$ level was the lowest (Figure 7c). Low corrected $J_{HONO}$ values were mostly associated
with low $OC/NO_3^-$ ratio, especially for aerosol samples collected in air polluted cities, such as Changji
and Xinxiang. However, Wangdu, a rural site in the North China Plain, where the $PM_{2.5}$ was dominated
by OM mainly due to local residential coal combustion (Liu et al., 2016; Li et al., 2024; Liu et al.,
2017), was an exception. As shown in Figure 5b, the $OC/NO_3^-$ ratio in clean days was generally higher
than that in polluted conditions. Interestingly, different from other cities, the $OC/NO_3^-$ ratio in
Guangzhou increased at polluted conditions, which was consistent with the correspondingly higher
corrected $J_{HONO}$ value. Guangzhou was located in the PRD region, and was characterized by large
fractions of OM in $PM_{2.5}$ due to large emission of VOCs from numerous manufacturing industries and
transport-related sources (Zheng et al., 2009), and the water-soluble organic carbon (WSOC) was the
dominated component in the organic aerosols (WSOC/OC=0.63) (Chang et al., 2019). It's reported that
organic compounds on the surface may act as photosensitizers in the photolysis of particulate nitrate
(Gen et al., 2022; Handley et al., 2007; Cao et al., 2022; Wang et al., 2021). The association of
particulate nitrate with organic matter may distort its molecular structure and enhance the absorption
cross section, resulting in significantly enhancement in the photochemical production of HONO. The
organic matter can also become hydrogen donors, and directly transfer hydrogen from organic
H-donors to $NO_2$ to form HONO (Gen et al., 2022). Therefore, we suggested that the gradually
increasing role of organic matter in $PM_{2.5}$ in China should be of great concern.
**3.3.3 Other factors**
The acidic proton may play an important role in the photochemical production of HONO
and affect the release of photolysis products (Bao et al., 2018; Scharko et al., 2014). Scharko et al.
(2014) found that gaseous HONO production from nitrate photolysis was the highest at the lowest



aerosol acidity (pH, ~2) and decreased with pH, and reached almost zero at pH higher than 4. In this
work, the estimated pH of these aerosol samples was in the range of 1.83−3.46 (the Extended Aerosol
Inorganic Model, E-AIM (Shi et al., 2021; Wexler and Clegg, 2002; Clegg et al., 1998)) with detailed
information provided in the Supporting Information. As shown in Figure S4, however, the correlation
between pH and $J_{HONO}$ was weak, which indicated that pH was an important factor, but not the key one
driving the spatial differences of $J_{HONO}$ in this work. Noting that halide ions, such as chlorine ($Cl^-$), may
lead to enhancement of surface nitrate anion and promote nitrate photolysis (Gen et al., 2022; Zhang et
al., 2020a), we also plotted $J_{HONO}$ against the molar ratio of $Cl^-$ to $NO_3^-$ ($Cl^-/NO_3^-$) in Figure 8a. Even
though Guangzhou was a southern coastal city, the sampling site in this work was far away from the
South China Sea (>50 km). Besides, during the observation period, the aerosol collected in Guangzhou
was more representative of inland aerosol instead of marine aerosol, with the air parcel usually coming
from inland directions (Figure 8b) and the ratio of $Cl^-$ to $NO_3^-$ (0.02) much lower than that in fresh sea
spray aerosol (>1.0) (Xiao et al., 2017; Pipalatkar et al., 2014; Atzei et al., 2019; Wang et al., 2019).
Therefore, we suggested that the halide ions were not the determining factor for the high $J_{HONO}$ value in
Guangzhou, and the exact role of halide ions in HONO formation through the photolysis of particulate
nitrate required further investigation.

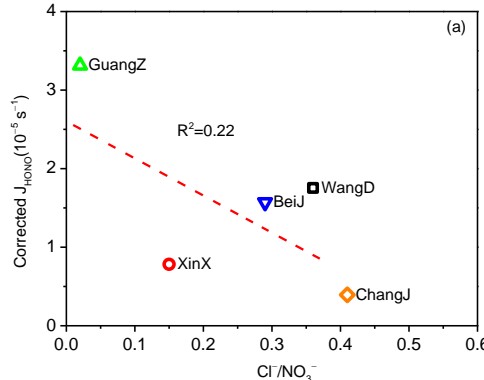


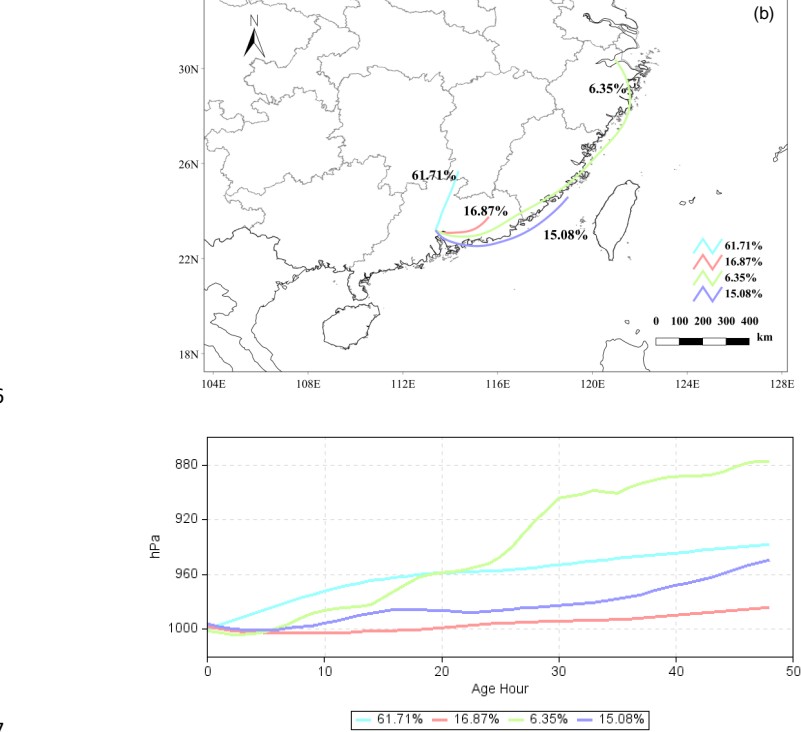



**Figure 8.** (a) Relationship between the average corrected $J_{HONO}$ and $Cl^-/NO_3^-$ under different sampling

locations, and (b) the back trajectory cluster analysis in Guangzhou during the sampling period.

**3.4 Environmental implication**

The determined $J_{HONO}$ was closely associated with the aerosol chemical and physical characteristics,

especially the coexisted organic components, and distributed around the curve as expressed by Eq. (6).

It's the first effort to explore the photolysis of particulate nitrate in aerosol samples collected from

different typical regions of China. The enhanced formation of HONO from the photolysis of particulate

nitrate can contribute significantly to the atmospheric oxidation capacity. To assess the photolysis of

particulate nitrate as a HONO daytime source, the noontime source strength of HONO ($S_{HONO}$) through

this mechanism in the air column within the planetary boundary layer can be calculated by the

following equation (Ye et al., 2017):

$$S_{HONO}\ (10^{-5}\ mol\ h^{-1} m^{-2}) = 0.67 \times NO_3^- (\mu mol\ m^{-3}) \times 10^{-6} \times J_{HONO} \times BLH \times 3600 \qquad (7)$$

or

$$S_{HONO}\ (ppbv\ h^{-1}) = 0.67 \times NO_3^- (ppbv) \times J_{HONO} \times 3600 \qquad (8)$$



392 where BLH means the boundary mixing height (m). Here, we assumed a typical BLH of 1000 m.

393 Based on the daily measured $NO_3^-$ and corrected $J_{HONO}$ value in each city, the $S_{HONO}$ derived from Eq.

394 (7) or (8) during the observation period was showed in Table 1. It was found that, even though the

395 $J_{HONO}$ in polluted days was much lower than that in clean days, due to the apparent higher $NO_3^-$

396 concentration, the corresponding $S_{HONO}$ was about twice the average in clean days. The calculated

397 $S_{HONO}$ ranged from $0.03 \times 10^{-5}$ mol $h^{-1}$ $m^{-2}$ to $0.88 \times 10^{-5}$ mol $h^{-1}$ $m^{-2}$ (0.01 ppbv $h^{-1}$−0.2 ppbv $h^{-1}$), with

398 the mean value of $0.36 \times 10^{-5}$ mol $h^{-1}$$m^{-2}$ (0.08 ppbv $h^{-1}$), which was comparable or higher than other

399 HONO sources (Bhattarai et al., 2019; Wang et al., 2023b; Ye et al., 2017). For example, the soil

400 HONO emission flux was measured in the range of $1.81 \times 10^{-6}$ mol $h^{-1}$ $m^{-2}$−$4.55 \times 10^{-6}$ mol $h^{-1}$ $m^{-2}$ in

401 the soil without suffering nitrogen fertilizer (Bhattarai et al., 2019). The mean value of $S_{HONO}$ during

402 the observation period was the highest in Wangdu ($0.42 \times 10^{-5}$ mol $h^{-1}$$m^{-2}$, 0.10 ppbv $h^{-1}$) and

403 Guangzhou ($0.41 \times 10^{-5}$ mol $h^{-1}$$m^{-2}$, 0.09 ppbv $h^{-1}$), followed by Xinxiang ($0.35 \times 10^{-5}$ mol $h^{-1}$$m^{-2}$, 0.08

404 ppbv $h^{-1}$), Beijing ($0.22 \times 10^{-5}$ mol $h^{-1}$$m^{-2}$, 0.05 ppbv $h^{-1}$), and Changji ($0.13 \times 10^{-5}$ mol $h^{-1}$$m^{-2}$, 0.03

405 ppbv $h^{-1}$). Even though the $PM_{2.5}$ and $NO_3^-$ concentration was the lowest in Guangzhou, the $S_{HONO}$ was

406 much higher than other cities with air pollution. It should be noted that the $S_{HONO}$ calculated with the

407 daily changed $NO_3^-$ and $J_{HONO}$ value in this work was much lower than the value reported by Bao et al.

408 (2018) (0.78 ppbv $h^{-1}$), which applied the average $NO_3^-$ (6.64 μg $m^{-3}$, 2.62 ppbv) and the $J_{HONO}$ range

409 ($1.22 \times 10^{-5}$ $s^{-1}$−$4.84 \times 10^{-4}$ $s^{-1}$) to simulate $S_{HONO}$ (0.12 ppbv $h^{-1}$−4.57 ppbv $h^{-1}$). Other works, such as Fu

410 et al. (2019) and Gu et al. (2022a), applied the mean value of $J_{HONO}$ ($8.3 \times 10^{-5}$ $s^{-1}$) and the observed

411 $NO_3^-$ concentration to calculate $S_{HONO}$. However, due to the significant decrease of $J_{HONO}$ along with

412 the increase of $NO_3^-$, the $S_{HONO}$ calculated with mean $NO_3^-$ or $J_{HONO}$ will be largely overestimated, thus

413 directly influencing the identification of HONO sources. For example, $J_{HONO}$ was the highest in

414 Wangdu in November 23, 2023 with the value of $19.6 \times 10^{-5}$ $s^{-1}$, while the corresponding $NO_3^-$

415 concentration was low (0.39 μg $m^{-3}$). If applying the average $NO_3^-$ concentration (12.53 μg $m^{-3}$,

416 equivalent to 4.53 ppbv) and the maximum $J_{HONO}$ value, the determined $S_{HONO}$ value would be

417 $9.56 \times 10^{-5}$ mol $h^{-1}$ $m^{-2}$ (2.14 ppbv $h^{-1}$), which was about 30 times higher than the actual result (0.07

418 ppbv $h^{-1}$). Therefore, we suggested to estimate $S_{HONO}$ with the observed concentration of $NO_3^-$ and the

419 $J_{HONO}$ value derived from the parameterization equation with $OC/NO_3^-$, thereby reducing the large

420 uncertainties and improving estimations of HONO budget.

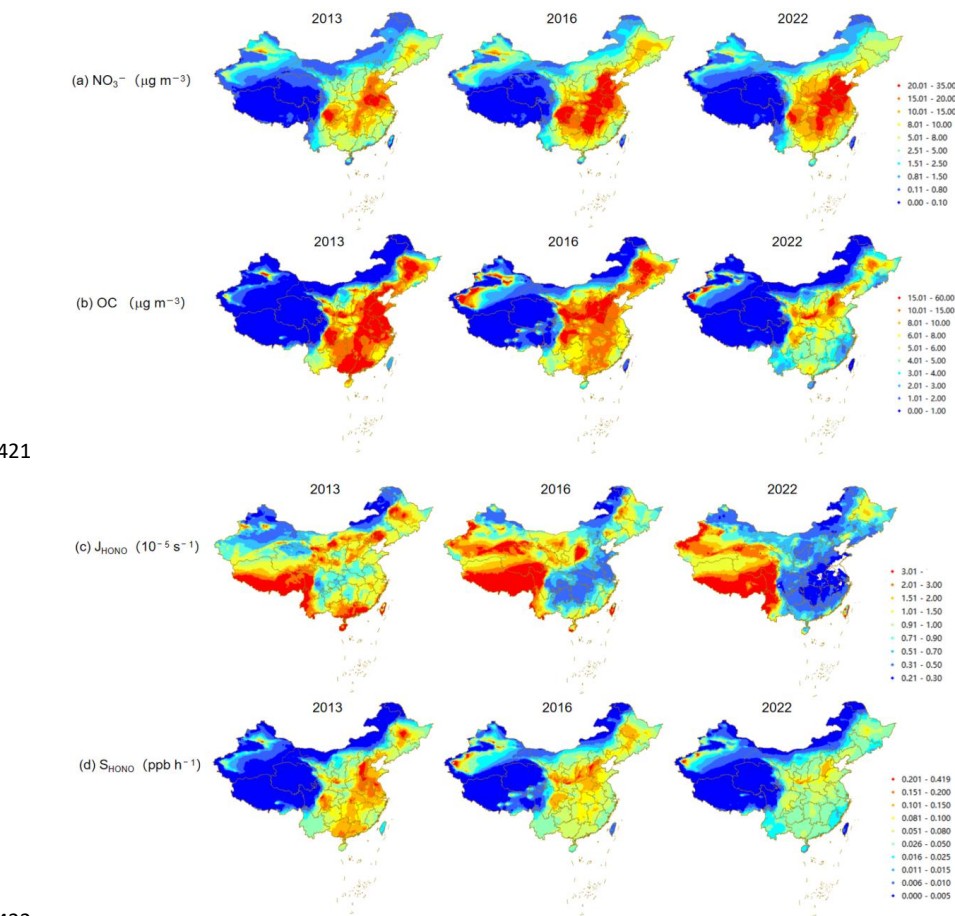

**Figure 9.** Spatial distributions of the average (a) $NO_3^-$, (b) OC, (c) $J_{HONO}$, and (d) $S_{HONO}$ from November 15 to December 15 in the year of 2013, 2016, and 2022 in China. The $J_{HONO}$ and $S_{HONO}$ estimated in this work were derived under the same environmental conditions (RH=65 %, temperature=20 ℃, and light intensity=150 kW $m^{-2}$), thus were more representative of the potential of HONO production rather than the actual value in the real ambient environment.

On the basis of the daily average concentrations of $NO_3^-$ and OC extracted from the Chinese high resolution $PM_{2.5}$ Component simulation concentration dataset (CAQRA-aerosol, https://www.capdatabase.cn, 15 km×15 km) (Kong, et al., 2024), the $J_{HONO}$ and $S_{HONO}$ can be estimated by Eq. (6) and (8), respectively. As shown in Figure 9, significant spatio-temporal change characteristics of $NO_3^-$, OC, $J_{HONO}$ and $S_{HONO}$ were demonstrated in autumn-winter seasons from 2013 to 2022 in China. The high $J_{HONO}$ were concentrated in the 'clean' environments (e.g., Tibetan Plateau



area, South Xinjiang Basin, Yunnan-Guizhou plateaus, and Sichuan basins) and followed by those air
polluted regions (e.g., NCP, Fenhe-Weihe Basin, Northeastern China, and PRD). From 2013 to 2022,
with OC decreasing significantly, while $NO_3^-$ keeping stable or even increasing, $J_{HONO}$ showed a
downward trend in most regions. Although the $J_{HONO}$ in polluted regions was comparatively lower than
that in 'clean' environments, the higher values of $S_{HONO}$ were mostly distributed in these polluted
regions resulting from the much higher $NO_3^-$ concentration. However, it should be noted that the
photolysis of particulate nitrate contributed only a small fraction to the needed daytime HONO source
in these polluted regions, such as 1.26−3. 82 ppbv h$^{-1}$ in the cities in the North China Plain (Hou et al.,
2016; Wang et al., 2017; Lian et al., 2022; Li et al., 2018), 0.75 ppbv h$^{-1}$ in the Western China (Huang
et al., 2017), and 0.77−4.90 ppbv h$^{-1}$ in Southern China (Li et al., 2012; Su et al., 2008). We noted that
uncertainties still exist in our simulations. Given the paucity of filed measurements of HONO
production from aerosol samples in 'clean' environments, the deviation of $J_{HONO}$ derived from the
parametrization in this work may be large in these regions. Additionally, the concentrations of $NO_3^-$
and OC extracted from the CAQRA-aerosol in 'clean' environments were around the mean deviation
level. Therefore, more field observations and simulation experiments should be taken in these 'clean'
regions in the future, to enrich and improve the parametric equations of $J_{HONO}$, and further evaluate the
contribution of nitrate photolysis to the formation of HONO in different regions in China.
**4 Conclusions**

This study for the first time systematically analyzed the production of HONO from the photolysis

of particulate nitrate in $PM_{2.5}$ samples from multiple sites across China, shedding light to the
contribution of this photolysis process to HONO daytime source in different environments. A total of
20 pairs of comparative photochemical experiments were conducted in Wangdu to evaluate and
quantify the shadowing effect. We found that the corrected $J_{HONO}$ values varied with sampling periods
and locations over a wide range, distributing from $0.16 \times 10^{-5}$ s$^{-1}$ to $19.60 \times 10^{-5}$ s$^{-1}$. The coexisted
organic components in $PM_{2.5}$ can promote the photolysis of particulate nitrate, with higher $J_{HONO}$
generally associated with higher $OC/NO_3^-$ ratio. Considering the logarithmical decrease of $J_{HONO}$ with
increased $NO_3^-$, we suggested that the $S_{HONO}$ should be calculated with $J_{HONO}$ derived from the
parameterization equation with $OC/NO_3^-$ instead of the average value. The photolysis of particulate



nitrate can become a potential daytime HONO source in southern urban cities, such as GuangZ, which was characterized by large VOCs emissions and enhanced formation of secondary particulate organic matter. Our work has provided an important reference for the research in other areas in the world with high proportion of organic components in aerosol samples, such as United States (Hass-Mitchell et al., 2024) and Europe (Bressi et al., 2021). To note, the filter samples collected in this work may not cover all representative environments in China, especially the background sites, more field observations and simulation experiments are needed in the future to better constrain the parameterization and mechanism of particulate nitrate photolysis.



**Data availability.** The data used in this paper can be provided upon request from the corresponding
author.

**Author contributions. J W, B L and K Z** conceived the study and designed the experiments. **J W, B**
**L, J G, C C, L W, Y Z, J L, Y Z, and X D** analyzed the data. **J W and B L** prepared the manuscript
and all the coauthors helped improve the manuscript.

**Competing interests.** The authors declare that they have no conflict of interest.

**Acknowledgement.** We thank the Data Integration Program of the Major Research Plan of the
National Natural Science Foundation of China (No. 92044303, https://www.capdatabase.cn) for
making the high-resolution simulation dataset of $PM_{2.5}$ chemical composition in Chinese from 2013 to
2020 available.

**Financial support.** This work was supported by the Central Level, Scientific Research Institutes for
Basic R&D Special Fund Business, China (No. 2022YSKY-26), and the National Key Research and
Development Program of China (No. 2022YFC3701100).



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
