# Peer review of "Exploring HONO production from particulate nitrate"

_EGUsphere, 2024_

## Author Comment (AC1)

A point-by-point response to the reviews

Thank you for your valuable comments. The followings are our responses to your comments.

**Response to Reviewer #1**

**Comment 1:** The authors of this manuscript estimated the photolysis rate constants of particulate nitrate for HONO production ($J_{NO_3^--HONO}$) through photolysis of $PM_{2.5}$ samples collected from five typical sites in China. They smartly revealed the "shadowing effect" of $PM_{2.5}$ filter samples on nitrate photolysis through investigating the difference of HONO production rates through photolysis of the $PM_{2.5}$ samples collected on a whole day and those sampled in both daytime and nighttime, finding that OC and EC played key roles in the "shadowing effect". Additionally, the authors further derived a parameterization equation of $J_{NO_3^--HONO}$ for atmospheric $PM_{2.5}$ based on significantly positive correlation between $J_{NO_3^--HONO}$ of $PM_{2.5}$ and $OC/NO_3^-$ ratio, which will be useful for precisely estimating $J_{NO_3^--HONO}$ in different areas with different aerosol chemical composition. In general, this manuscript is well organized, containing useful information about daytime HONO source from photolysis of atmospheric $PM_{2.5}$. This reviewer recommends the manuscript to be published in the journal.

**Answer:** Thank you for your approval. According to your valuable comments, we have made corresponding revisions in our revised manuscript.

**Comment 2:** The symbol used for indicating photolysis rate constant of particulate nitrate for HONO production is suggested to be $J_{NO_3^--HONO}$, rather than $J_{HONO}$ because $J_{HONO}$ is prevailingly adopted to represent the photolysis rate constant of HONO.

**Answer:** Thank you for your valuable comments. $J_{HONO}$ has been replaced by $J_{NO_3^--HONO}$ in our revised manuscript to represent the photolysis rate constant of particulate nitrate for HONO production.

**Comment 3:** The information of the five sampling sites are suggested to cite corresponding references.

**Answer:** The corresponding references of the five sampling sites have been added in the Supporting Information (S1.1 Sampling sites).

**Comment 4:** The derived $J_{NO_3^- - HONO}$ values strongly depended on the irradiation time, light intensity and RH according to Eq.(1)-(2) and Fig. 2, and thus it is better to mention about the key information for comparison of the J values with previous studies, e.g., the experiments with irradiation time of ~10 min for Ye et al. (2017), 15min for Bao et al. (2018), whereas 20 min for your experiments. Additionally, the J values derived by Ye et al. (2017) were based on production of the sum of HONO and $NO_2$.

**Answer:** Thank you for your valuable comments. According to your suggestions, we have added the information of the experimental conditions, such as irradiation time, temperature and RH, when comparing the derived $J_{NO_3^- - HONO}$ values with previous studies.( Page 12, line 261–264)

**Comment 5:** Did you measure the particulate nitrate concentration after the irradiation? How much did the formed HONO account for the consumed nitrate?

**Answer:** We had not measured the particulate nitrate concentration after the irradiation. Ye et al. (2017) has conducted an experiment to compare the amounts of particulate nitrate on two halves of a filter sample, one half undergone the light-exposure and the other half kept in freezer. The difference in the determined amounts of particulate nitrate between these two half filters was well predicted by HONO and $NO_2$ production in the light-exposure experiment, with an error less than 10%.

**Comment 6:** Lines 173-174, the description of "the consumption of reactive electron donors, such as acidic proton" is not correct because acidic proton is a proton donor, rather than a reactive electron donor.

**Answer:** The work by Bao et al. (2018) found that decrease in the HONO production was caused by consumption of reactive electron donors through Eq. (1). The mistake has been corrected in our revised manuscript:
" Previous works have revealed that the decay of HONO generation during light exposure period was not resulted from the evaporation loss of particulate nitrate (Ye et al., 2017), but mainly related to the inhomogeneity of particulate nitrate photochemical reactivity or the consumption of reactive electron donors." (Page 8, line 175)

$$HNO_3 + 2H^+ + 2e^- + hv \rightarrow HONO + H_2O \tag{1}$$

**Comment 7:** Lines 190-192, "$P_{HONO}$ does not increase" should be "$P_{HONO}$ didn't increase".

Answer: "$P_{HONO}$ does not increase"→"$P_{HONO}$ did not increase".(Page 9, line 192)

**Comment 8:** Line 207, the meaning of the second item in Eq. (4) is not clear.

Answer: Thank you for valuable comments. $P_{observed}^{HONO}$ represented the observed production rate of HONO from particulate nitrate photolysis through photochemical experiment, and $P_{corrected}^{HONO}$ represented the corrected value of $P_{HONO}$ after eliminating the shadowing effect. The above description has been added in our revised manuscript. (Page 10, line 210–212)

**Comment 9:** Lines 271-273, the values of various parameters for clean and polluted should present in a range, rather than fix values, or you have to mention the representatives of the values, e.g., the mean or average.

Answer: Thank you for valuable comments. The values of various parameters for clean and polluted conditions have been presented in a range ( mean ± 1SD) in Table R1.

**Table R1.** The concentrations of $PM_{2.5}$, $NO_3^-$, and OC, $OC/NO_3^-$, corrected $J_{NO_3^- - HONO}$, and $S_{HONO}$ in five representative cities in China under different air conditions during the sampling period.

| Site | Air condition | $PM_{2.5}$ ($\mu g\ m^{-3}$) | $NO_3^-$ ($\mu g\ m^{-3}$) | OC ($\mu g\ m^{-3}$) | $OC/NO_3^-$ | Corrected $J_{NO_3^- - HONO}$ ($10^{-5}\ s^{-1}$) [a] | $S_{HONO}$ ($10^{-5}$ mol $h^{-1}\ m^{-2}$) [b] | $S_{HONO}$ (ppbv $h^{-1}$) [c] |
|---|---|---|---|---|---|---|---|---|
| Beijing | Clean | 19.71±8.65 | 3.15±2.34 | 3.89±2.13 | 2.25±3.03 | 2.01±2.44 | 0.15±0.07 | 0.03±0.02 |
| | Polluted | 72.56±23.78 | 19.71±10.72 | 12.62±2.18 | 0.87±0.62 | 0.61±0.30 | 0.38±0.11 | 0.09±0.02 |
| | Whole-Min | 4.32 | 0.08 | 1.07 | 0.32 | 0.21 | 0.04 | 0.01 |
| | Whole-Max | 102.64 | 32.90 | 15.95 | 12.82 | 11.06 | 0.57 | 0.13 |
| | Whole-Mean | 32.92 | 7.29 | 6.07 | 1.85 | 1.57 | 0.22 | 0.05 |
| Changji | Clean | 20.39±6.00 | 3.05±1.75 | 3.61±1.08 | 1.66±1.11 | 0.65±0.18 | 0.07±0.03 | 0.02±0.01 |
| | Polluted | 80.49±39.54 | 20.59±4.74 | 8.35±2.97 | 0.44±0.08 | 0.21±0.03 | 0.16±0.04 | 0.04±0.01 |
| | Whole-Min | 14.45 | 0.88 | 2.69 | 0.28 | 0.16 | 0.03 [d] | 0.01 [d] |
| | Whole-Max | 169.35 | 28.28 | 14.34 | 3.65 | 0.91 | 0.22 | 0.05 |
| | Whole-Mean | 57.37 | 13.84 | 6.53 | 0.91 | 0.39 | 0.13 | 0.03 |
| Guangzhou | Clean | 25.62±6.08 | 3.29±1.68 | 6.89±2.21 | 2.72±1.79 | 3.25±1.28 | 0.36±0.15 | 0.08±0.03 |
| | Polluted | 40.32±2.23 | 4.38±1.30 | 13.82±1.34 | 3.35±0.86 | 3.53±0.61 | 0.59±0.15 | 0.13±0.03 |
| | Whole-Min | 14.77 | 0.85 | 3.67 | 0.82 | 1.37 | 0.17 | 0.04 |
| | Whole-Max | 42.74 | 6.63 | 15.62 | 8.05 | 5.83 | 0.75 | 0.17 |
| | Whole-Mean | 29.12 | 3.55 | 8.54 | 2.87 | 3.31 | 0.41 | 0.09 |
| Wangdu | Clean | 22.16±7.66 | 3.29±2.59 | 5.36±2.38 | 4.79±6.46 | 3.80±5.10 | 0.20±0.09 | 0.04±0.02 |
| | Polluted | 83.53±30.47 | 18.06±12.48 | 23.23±9.62 | 1.88±1.67 | 1.09±0.87 | 0.50±0.15 | 0.11±0.03 |
| | Whole-Min | 10.67 | 0.24 | 2.72 | 0.22 | 0.23 | 0.06 | 0.01 |
| | Whole-Max | 173.45 | 60.28 | 63.07 | 22.06 | 19.60 | 0.88 [e] | 0.20 [e] |
| | Whole-Mean | 68.38 | 14.41 | 18.82 | 2.60 | 1.75 | 0.42 | 0.10 |

| | | | | | | | | |
|---|---|---|---|---|---|---|---|---|
| | Clean | 23.53±5.45 | 4.35±1.41 | 5.69±2.46 | 1.37±0.61 | 1.28±0.49 | 0.21±0.07 | 0.05±0.02 |
| | Polluted | 68.98±33.43 | 24.87±21.5 | 14.63±4.41 | 0.87±0.45 | 0.62±0.35 | 0.40±0.12 | 0.09±0.03 |
| Xinxiang | Whole-Min | 18.32 | 2.37 | 2.33 | 0.30 | 0.19 | 0.09 | 0.02 |
| | Whole-Max | 143.10 | 73.47 | 22.06 | 2.02 | 1.96 | 0.59 | 0.13 |
| | Whole-Mean | 57.62 | 19.74 | 12.40 | 0.99 | 0.78 | 0.35 | 0.08 |

[a] represented the photolysis rate constant of particulate nitrate leading to HONO production after considering the influence of the shadowing effect. [b, c] represented the noontime source strength of HONO through the photolysis of particulate nitrate with the units of $10^{-5}$ mol h$^{-1}$ m$^{-2}$ and ppbv h$^{-1}$, respectively.[d, e] represented the minimum and maximum values of $S_{HONO}$ during the observation period.

**References**

Bao, F., Li, M., Zhang, Y., Chen, C., and Zhao, J.: Photochemical aging of Beijing urban $PM_{2.5}$: HONO production, Environ. Sci. Technol., 52, 6309-6316, 10.1021/acs.est.8b00538, 2018.

Ye, C., Zhang, N., Gao, H., and Zhou, X.: Photolysis of particulate nitrate as a source of HONO and NOx, Environ. Sci. Technol., 51, 6849-6856, 10.1021/acs.est.7b00387, 2017.

---

## Author Comment (AC2)

A point-by-point response to the reviews

Thank you for your valuable comments. The followings are our responses to your comments.

**Response to Reviewer #2**

**Comment 1:** This work studied HONO production from particulate nitrate photolysis in $PM_{2.5}$ at five Chinese cities. The method is well-designed and the shadowing effect of the artificial filter samples is corrected to some degree. This work provided good evidence to show the important role of $OC/NO_3^-$ in determining the $JNO_3^-$. The parameterization of corrected $JNO_3^-$ using the $OC/NO_3^-$ could improve the atmospheric relevance of laboratory studies conducted with aerosol filter samples and better quantify of HONO production from the photochemical reduction of nitrate. In general, this work is of high quality and significant importance to atmospheric chemistry and physics. I recommend the manuscript be accepted for publication after minor revisions.

**Answer:** Thank you for your approval. According to your valuable comments, we have made corresponding revisions in our revised manuscript.

**Comment 2:** $J_{HONO}$ can be confused with the photolysis of HONO. I suggest using "$J_{NO_3^- - HONO}$" instead.

**Answer:** Thank you for your valuable comments. $J_{HONO}$ has been replaced by $J_{NO_3^- - HONO}$ in our revised manuscript to represent the photolysis rate constant of particulate nitrate for HONO production.

**Comment 3:** I suggest $NO_3^-$ be considered and normalized while studying pH influence.

**Answer:** Thank you for your valuable comments. According to your suggestions, we considered $NO_3^-$ when studying the influence of pH on $J_{NO_3^- - HONO}$. As shown in Figure S1, the correlation between $J_{NO_3^- - HONO}$ and $pH/NO_3^-$ has enhanced comparing with that between $J_{NO_3^- - HONO}$ and pH, however this relationship was still weak ($R^2$=0.15), which indicated that pH was an important factor, but not the key one driving the spatial differences of $J_{NO_3^- - HONO}$ in this work. This conclusion was the same as before.

In general, there was no significant relationship between pH and $NO_3^-$ for $PM_{2.5}$ samples, thus there is no need to eliminate the influence of $NO_3^-$. Besides, previous works generally considered pH instead of $pH/NO_3^-$ when studying the influence of pH (Scharko et al., 2014). Thus, for better comparison with previous studies, we recommended to use a uniform expression and use pH instead of $pH/NO_3^-$ in this work when exploring the influence of pH on $J_{NO_3^- - HONO}$.

[Figure]

**Figure S1.** Relationships between the average corrected $J_{NO_3^- - HONO}$ and $pH/NO_3^-$ under different sampling locations.

**Comment 4:** Line 17, the light screening coefficient needs to be determined before use.

**Answer:** Thank you for your suggestions. The content in abstract has been reorganized and revised in our manuscript (Page 1, line 16–19):
"We developed a method to correct and quantify the 'shadowing effect'— potential light extinction within aerosol layers at heavy $PM_{2.5}$ loadings on the filters—for $J_{NO_3^- - HONO}$ measurements, which showing that elemental carbon (EC), the dominant light-absorbing component in $PM_{2.5}$, played a dominant role in it."

**Comment 5:** Line 17, EC may be determined before use.

**Answer:** The content in abstract has been reorganized and revised in our manuscript (Page 1, line 16–19):
"We developed a method to correct and quantify the 'shadowing effect'— potential light extinction within aerosol layers at heavy $PM_{2.5}$ loadings on the filters—for $J_{NO_3^- - HONO}$ measurements, which showing that elemental carbon (EC), the dominant light-absorbing component in $PM_{2.5}$, played a dominant role in it."

**Comment 6:** Line 18, what is "corrected" $J_{HONO}$?

**Answer:** "Corrected $J_{NO_3^- - HONO}$" represented the corrected value of $J_{NO_3^- - HONO}$ after considering the influence of the "shadowing effect".

**Comment 7:** Line 25-28, I am confused by the sentence, is the contribution of photolysis of nitrate potentially important or limited? and why?

**Answer:** In this work, the calculated noontime source strength of HONO ($S_{HONO}$) through the photolysis of particulate nitrate ranged from $0.03 \times 10^{-5}$ mol $h^{-1}$ $m^{-2}$ to $0.88 \times 10^{-5}$ mol $h^{-1}$ $m^{-2}$ (0.01 ppbv $h^{-1}$−0.2 ppbv $h^{-1}$), with the mean value of $0.36 \times 10^{-5}$ mol $h^{-1}$$m^{-2}$ (0.08 ppbv $h^{-1}$), which was comparable or higher than other HONO sources (Bhattarai et al., 2019; Wang et al., 2023; Ye et al., 2017). For example, the soil HONO emission flux was measured in the range of $1.81 \times 10^{-6}$ mol $h^{-1}$ $m^{-2}$−$4.55 \times 10^{-6}$ mol $h^{-1}$ $m^{-2}$ in the soil without suffering nitrogen fertilizer (Bhattarai et al., 2019). However, it should be noted that the photolysis of particulate nitrate contributed only a small fraction to the reported daytime HONO unknown source in these regions, such as 1.26−3. 82 ppbv $h^{-1}$ in the cities in the North China Plain (Hou et al., 2016; Wang et al., 2017; Lian et al., 2022; Li et al., 2018), 0.75 ppbv $h^{-1}$ in the Western China (Huang et al., 2017), and 0.77−4.90 ppbv $h^{-1}$ in Southern China (Li et al., 2012; Su et al., 2008). Therefore, the contribution of the photolysis of particulate nitrate to daytime HONO unknown source was still limited, and other sources were still needed to account for the daytime unknown HONO source in these regions. To avoid misunderstandings, this sentence has been revised in our manuscript (Page 1, line 26−28):
"This study confirms that the photolysis of particulate nitrate can be a potential HONO daytime source in rural or southern urban sites, which are characterized by high proportion of organic matter in $PM_{2.5}$."

**Comment 8:** Line 67-69, How do mechanisms and dominant factors contribute to accurately estimating contributions?

**Answer:** The rate constant of nitrate photolysis ($J_{NO_3^- - HONO}$) showed large variability in different environments. In New York, Ye et al. (2017) reported that the photolysis rates of particulate nitrate in clean areas were two orders of magnitude higher than that in polluted areas, ranging from $6.2 \times 10^{-6}$ to $5.0 \times 10^{-4}$ $s^{-1}$. The large discrepancies may undoubtedly impacted the estimation of HONO production rates from nitrate photolysis. This highly-varied value of $J_{NO_3^- - HONO}$ was closely associated with environmental conditions and the aerosol chemical or physical characteristics, such as relative humidity (RH), aerosol acidity, light intensity, and coexisting components

(organic components, halogen, etc.). Thus, elucidating the mechanism and dominant factors controlling the photolysis of particulate nitrate can reduce the uncertainty in $J_{NO_3^- - HONO}$ predictions and improve estimations of the source strength of HONO production from this process. This sentence has been revised in our manuscript (Page 4, line 68–70):

"Elucidating the mechanism and dominant factors controlling the photolysis of particulate nitrate is important to accurately estimate the HONO production rates from nitrate photolysis, thus improving estimations of HONO budgets."

**Comment 9:** Line 74, why is the shadowing effect significant only under haze conditions? Are the authors assuming that all samples were collected within 24 hours? is this true?

**Answer:** Yes. Generally, we assumed that the sampling time of all the aerosol samples were the same. The $PM_{2.5}$ loading on the filters collected under haze conditions was much higher than that under clean conditions, thus showing much significant shadowing effect.

**Comment 10:** Line 80, $NO_3^-$ changed over time or location?

**Answer:** $NO_3^-$ change over locations. This sentence has been revised in our manuscript (Page 4, line 80–81):
"According to previous field observations, the $PM_{2.5}$ chemical composition, especially particulate nitrate ($NO_3^-$), showed obvious spatial differences across China."

**Comment 11:** Line 190, I understand that the shadowing effect causes a lower j at higher nitrate loading. How could it lead to a lower P at higher nitrate loading?

**Answer:** Thank you for your valuable comments. It is noted that this phenomenon— lower P at higher nitrate loading—has also been observed in other works (Ye et al., 2017; Bao et al., 2018).We assumed that more than single layer of $PM_{2.5}$ particles were collected on the filter samples at higher $PM_{2.5}$ loading. Light absorption components in $PM_{2.5}$, such as EC, may prevent the UV light transmitting efficiently through the upper layers. The particulate nitrate underneath the aerosol filters may receive less UV light, thus inhibiting the photolysis of particulate nitrate. Thus, the increase of light absorption components in $PM_{2.5}$ at very high $NO_3^-$ loading condition may hinder the light absorption of nitrate inside and underside of the particle, which may not only lead to the decrease of the upward trend of $P_{HONO}$ but also a lower value at higher nitrate loading.

**Comment 12:** Line 195: should the underestimation of the p or J value primarily depend on the loading time or the amount of light-absorbing species? Is it not necessarily related to whether the environment is polluted or not?

**Answer:** The underestimation of the $P_{HONO}$ or $J_{NO_3^- - HONO}$ value primarily depended on the amount of light-absorbing species on the filters. But the the amount of light-absorbing species was related to the sampling time and air conditions. Generally, we assumed that the sampling time of all the aerosol filter samples were the same. Thus, the $PM_{2.5}$ loading on the filters collected under polluted conditions was much higher than that under clean conditions, thus showing much significant shadowing effect.

**Comment 13:** Line 337, I can hardly see a clear dependence of $OC/NO_3^-$ on $PM_{2.5}$ levels.

**Answer:** As can be seen in Figure 7c, high $OC/NO_3^-$ ratios generally happened in clean areas, such as Guangzhou, where the averaged $PM_{2.5}$ level was the lowest among these cites. Cities with high $PM_{2.5}$ levels generally have low ratios of $OC/NO_3^-$, such as Changji and Xinxiang, however, there was an exception—Wangdu, a rural site in the North China Plain, where the $PM_{2.5}$ was high but dominated by OM mainly due to local residential coal combustion. The expression has been revised in our manuscript (Page 17, line 345−348):
"Generally, cities with higher $PM_{2.5}$ levels have lower $OC/NO_3^-$ ratios, such as Changji and Xinxiang, however, there was an exception—Wangdu, a rural site in the North China Plain, where the $PM_{2.5}$ was high but dominated by OM mainly due to local residential coal combustion."

**Comment 14:** More details should be added to the caption of Fig. 9, such as an explanation of how these values were obtained.

**Answer:** The explanation of how these values were obtained has bee added in our revised manuscript (Page 22, line 433−435):
" The daily average concentrations of $NO_3^-$ and OC were extracted from the Chinese high resolution $PM_{2.5}$ Component simulation concentration dataset (Kong, et al., 2024). "

**References**

Bao, F., Li, M., Zhang, Y., Chen, C., and Zhao, J.: Photochemical aging of Beijing urban $PM_{2.5}$: HONO production, Environ. Sci. Technol., 52, 6309-6316, 10.1021/acs.est.8b00538, 2018.

Bhattarai, H. R., Liimatainen, M., Nykänen, H., Kivimäenpää, M., Martikainen, P. J., and Maljanen, M.: Germinating wheat promotes the emission of atmospherically significant nitrous acid (HONO) gas from soils, Soil Biol. Biochem., 136, 10.1016/j.soilbio.2019.06.014, 2019.

Hou, S., Tong, S., Ge, M., and An, J.: Comparison of atmospheric nitrous acid during severe haze and clean periods in Beijing, China, Atmos. Environ., 124, 199-206, 10.1016/j.atmosenv.2015.06.023, 2016.

Huang, R.-J., Yang, L., Cao, J., Wang, Q., Tie, X., Ho, K.-F., Shen, Z., Zhang, R., Li, G., Zhu, C., Zhang, N., Dai, W., Zhou, J., Liu, S., Chen, Y., Chen, J., and O'Dowd, C. D.: Concentration and sources of atmospheric nitrous acid (HONO) at an urban site in Western China, Sci. Total Environ., 593-594, 165-172, https://doi.org/10.1016/j.scitotenv.2017.02.166, 2017.

Kong, L., Tang, X., Zhu, J., Wang, Z., Liu, B., Zhu, Y., Zhu, L., Chen, D., Hu, K., Wu, H., Wu, Q., Shen, J., Sun, Y., Liu, Z., Xin, J., Ji, D., and Zheng, M.: High-resolution simulation dataset of hourly $PM_{2.5}$ chemical composition in China (CAQRA-aerosol) from 2013 to 2020, Adv. Atmos. Sci., 41, 1-16, 10.1007/s00376-024-4046-5, 2024.

Li, D., Xue, L., Wen, L., Wang, X., Chen, T., Mellouki, A., Chen, J., and Wang, W.: Characteristics and sources of nitrous acid in an urban atmosphere of northern China: Results from 1-yr continuous observations, Atmos. Environ., 182, 296-306, https://doi.org/10.1016/j.atmosenv.2018.03.033, 2018.

Li, X., Brauers, T., Häseler, R., Bohn, B., Fuchs, H., Hofzumahaus, A., Holland, F., Lou, S., Lu, K. D., Rohrer, F., Hu, M., Zeng, L. M., Zhang, Y. H., Garland, R. M., Su, H., Nowak, A., Wiedensohler, A., Takegawa, N., Shao, M., and Wahner, A.: Exploring the atmospheric chemistry of nitrous acid (HONO) at a rural site in Southern China, Atmos. Chem. Phys., 12, 1497-1513, 10.5194/acp-12-1497-2012, 2012.

Lian, C., Wang, W., Chen, Y., Zhang, Y., Zhang, J., Liu, Y., Fan, X., Li, C., Zhan, J., Lin, Z., Hua, C., Zhang, W., Liu, M., Li, J., Wang, X., An, J., and Ge, M.: Long-term winter observation of nitrous acid in the urban area of Beijing, J. Environ. Sci. (China), 114, 334-342, 10.1016/j.jes.2021.09.010, 2022.

Scharko, N. K., Berke, A. E., and Raff, J. D.: Release of nitrous acid and nitrogen dioxide from nitrate photolysis in acidic aqueous solutions, Environ. Sci. Technol., 48, 11991-12001, 10.1021/es503088x, 2014.

Su, H., Cheng, Y. F., Shao, M., Gao, D. F., Yu, Z. Y., Zeng, L. M., Slanina, J., Zhang, Y. H., and Wiedensohler, A.: Nitrous acid (HONO) and its daytime sources at a rural site during the 2004 PRIDE-PRD experiment in China, J. Geophys. Res. Atmos., 113, 10.1029/2007jd009060, 2008.

Wang, J., Zhang, X., Guo, J., Wang, Z., and Zhang, M.: Observation of nitrous acid (HONO) in Beijing, China: Seasonal variation, nocturnal formation and daytime budget, Sci. Total Environ., 587-588, 350-359, 10.1016/j.scitotenv.2017.02.159, 2017.

Wang, Y., Fu, X., Wang, T., Ma, J., Gao, H., Wang, X., and Pu, W.: Large contribution of nitrous acid to soil-emitted reactive oxidized nitrogen and its effect on air quality, Environ. Sci. Technol., 57, 3516-3526, 10.1021/acs.est.2c07793, 2023.

Ye, C., Zhang, N., Gao, H., and Zhou, X.: Photolysis of particulate nitrate as a source of HONO and NOx, Environ. Sci. Technol., 51, 6849-6856, 10.1021/acs.est.7b00387, 2017.

---

## Author Response (AR2)

Dear Professor Benjamin A Nault:

Thank you very much for handling our manuscript submitted to ***Atmospheric Chemistry and Physics*** (**MS No.:** egusphere-2024-2141; **Title:** Exploring HONO production from particulate nitrate photolysis in Chinese representative regions: characteristics, influencing factors and environmental implications).

We deeply thank you and two reviewers for giving constructive comments and suggestions that are very helpful to improve our manuscript. According to your valuable suggestions, we have included a response to Reviewer #2, Comments 9, 11, and 12 into our revised manuscript to ensure readers understand the assumptions/rational used in this manuscript (Page 4, Line 77-80; Page 9, Line 194-200). We hope the revised manuscript meet the publication standards.

On behalf of all the co-authors, I would like to thank you and referees for all the invaluable comments. Please feel free to contact me if you need any further information.

Yours Sincerely,

Dr. Jiaqi Wang

Chinese Research Academy of Environmental Sciences, Beijing 100012, China

E-mail address: wang.jiaqi@craes.org.cn

Phone: 18760106801